



# An engineering model for 3D turbulent wind inflow based on a limited set of random variables

Manuel Fluck[1] and Curran Crawford[1]

[1]Department of Mechanical Engineering, Institute for Integrated Energy Systems (IESVic), University of Victoria, Victoria, BC, Canada

*Correspondence to:* Manuel Fluck (mfluck@uvic.ca)

**Abstract.** Emerging stochastic analysis methods are of potentially great benefit for wind turbine power output and loads analysis. Instead of requiring multiple (e.g. ten-minute) deterministic simulations, a stochastic approach can enable quick assessment of a turbine's long term performance (e.g. 20 year fatigue and extreme loads) from a single stochastic simulation. However, even though the wind inflow is often described as a stochastic process, the common spectral formulation requires a
5  large number of random variables to be considered. This is a major issue for stochastic methods, which suffer from the 'curse of dimensionality' leading to a steep performance drop with an increasing number of random variables contained in the governing equations. In this paper a novel engineering wind model is developed which reduces the number of random variables by 4–5 orders of magnitude compared to typical models while retaining proper spatial correlation of wind speed sample points across a wind turbine rotor. The new model can then be used as input to direct stochastic simulations models under development. A
10  comparison of the new method to results from the commercial code *TurbSim* and a custom implementation of the standard spectral model shows that for a 3D wind field the most important properties (cross-correlation, covariance, auto- and cross-spectrum) are conserved adequately by the proposed method.

**Nomenclature**

Latin Letters:

| | |
|---|---|
| $e$ | Euler's number |
| $\boldsymbol{f} = [f_m]$ | frequency |
| $i = \sqrt{-1}$ | imaginary unit |
| $N_F$ | number of frequencies |
| $N_P$ | number of wind speed points |
| $N_R$ | number of random variables |
| $P$ | a point in Euclidean space |
| $S_{kk}$ | (auto) power spectrum |
| $S_{kj}$ | cross power spectrum |
| $t$ | time |



| $U$ | wind speed Fourier coefficient |
| $u$ | wind speed velocity |

Greek Letters:

| $\theta$ | phase angle |
| $\Delta\theta$ | phase angle increment |
| $\xi$ | random number |
| $\boldsymbol{\omega} = [\omega_m]$ | angular frequency |

Indices:

| $j, k$ | points in space |
| $m$ | frequencies |

# 1   Introduction

Engineering design tasks frequently face uncertain or random model parameters (e.g. imprecise component geometries), system properties (e.g. tolerances on manufacturing quality), and/ or boundary conditions (e.g. varying wind conditions). In a deterministic modeling framework the analysis of such uncertain systems produces one specific solution for each realization

of the random quantity. A 'realization' (also referred to as one 'sample') is one specific observation of the random quantity, for example a specific solution for one specific geometry, or one specific set (a realization) of inflow conditions. In a numerical experiment a realization is usually obtained base on the generation of one specific random seed. However, through this process the stochastic dimension of the problem at hand is either ignored entirely, by analyzing the most likely case only (the purely deterministic approach), or it requires multiple parallel solutions to asses the statistics of the results a posteriori, for example

via extreme value, sensitivity analysis, or Monte Carlo simulation. Often the first two options are insufficient, and the latter is computationally too expensive. To solve this dilemma the focus of recent research has lately moved towards stochastic analyses and uncertainty quantification (Sudret, 2007; Najm, 2009; Le Maître and Knio, 2010; Sullivan, 2015). Rather than generating one specific solution for each realization of a random input or model quantity, a stochastic analysis can help assess uncertainties quicker and even include uncertain quantities directly into the system analysis. This is because it not only provides one specific

solution, but it solves the problem for the whole ensemble of all possible realizations at once, see Fig. 1. This is made possible via stochastic methods that transform the problem from multiple deterministic realizations with random seeds to a formulation of the govening equations that directly describe the stochastic variables in the system. The stochastic solution then directly describes the statistics (e.g. the probability distributions) of the outputs, based on the properties given for the input variables in the forcing terms of the governing system equations.

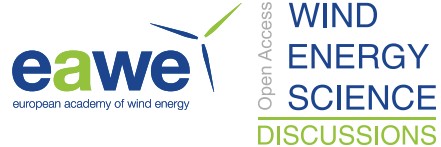

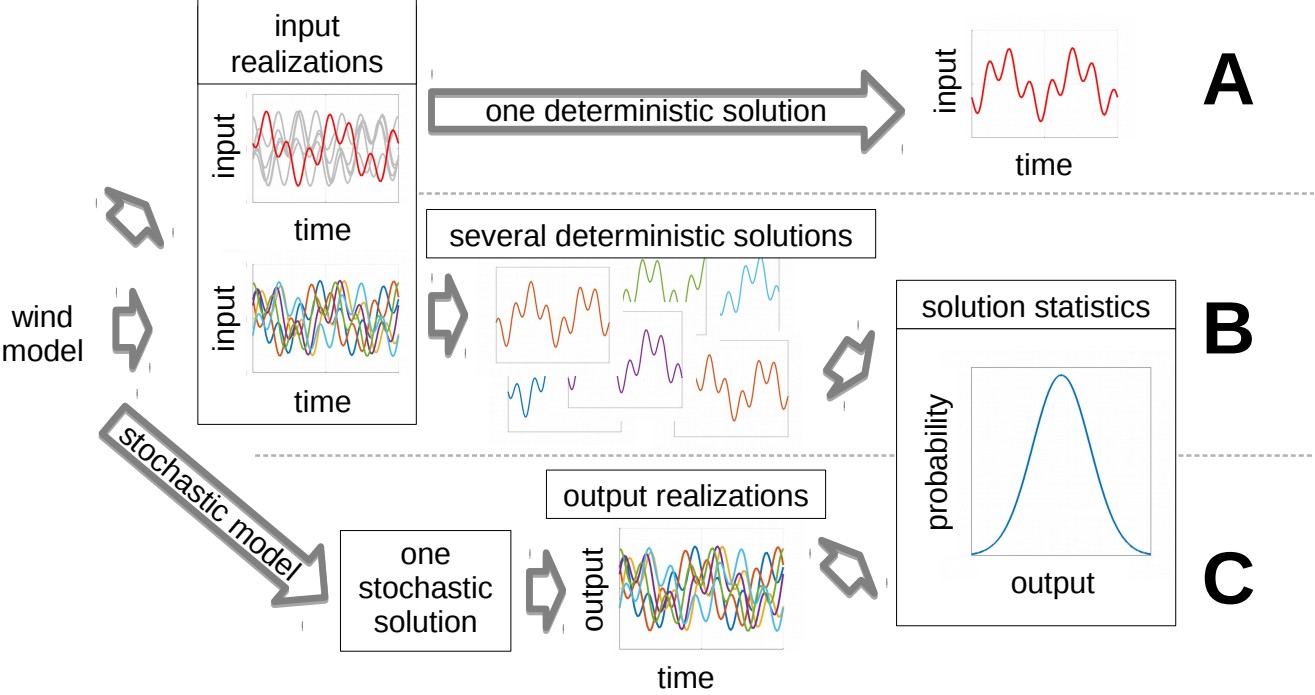

**Figure 1.** Comparison of the solution processes in a pure deterministic (A), a deterministic-statistic (B), and a stochastic framework (C).

In wind turbine engineering, the driving force, the turbulent atmospheric wind, is commonly described as a stochastic field, derived from turbulent wind models developed around stochastic ten-minute mean wind speed distributions. This naturally invites the use of stochastic methods to asses extreme and fatigue loads, annual power production, power fluctuations, etc. in a stochastic sense and thus exploit the advantages of stochastic methods. However, wind turbine design and analysis is usually

carried out in a deterministic fashion, or at best as a Monte-Carlo-like set of several subsequent deterministic solutions (path A and B in Fig. 1 respectively). The wind turbine design standard IEC 61400-1, Ed. 3 (2005) is indicative of this deterministic framework. It bases the turbine load analysis on multiple deterministic simulations, carried out at many different mean wind speeds, for about 20 different load cases, each simulated for ten minutes and repeated several times with different realizations of the turbulent inflow, each generated from a different numerical random seed. For a land based turbine this quickly amounts

to evaluating several hundred ten-minute samples. For offshore turbines, where various wind conditions (wind speed and direction) additionally have to be combined with various sea states (combinations of wave height and direction) this number increases to several thousand ten-minute evaluations. However, even with a large number of deterministic simulations, extrapolation to extreme loads is a delicate exercise and results can vary greatly (Moriarty, 2008; Burton et al., 2011). Moreover, Zwick and Muskulus (2015) show that basing a wind turbine analysis on six ten-minute wind speed simulations, generated

from six different random seeds, results in a difference of up to 34% in the ultimate loads for the most extreme 1% of seed combinations. Tibaldi et al. (2014) present a study, which indicates that turbine loads extracted even from 20 different ten-




minute wind fields, generated from 20 different random seeds, vary greatly. This shows that in a deterministic framework load variations from different random seeds can dominate effects from design parameter changes even with a fairly large number of realizations analyzed. Obviously this constitutes a severe problem, particularly when concerned with gradient-based optimization where not only relatively fast solutions times, but also reliable design variable gradients are vital.

A direct stochastic treatment of the wind loading (path C in Fig. 1), on the other hand, considers the wind as a stochastic process throughout the turbine simulation procedure. It postpones the generation of realizations until after the calculation of a solution for the system equations, which thus become stochastic equations. Hence it can be a means to efficiently include stochastic parameters, directly obtain a stochastic solution, and arrive at the statistics of the resulting loads much quicker. Fluck and Crawford (2016a) present an example of a stochastic analysis for wing loads in turbulent inflow, and show that such a stochastic approach does not rely on the repeated analysis of multiple (e.g. 600 s) realizations of the wind field. Instead one (possibly short, e.g. 10 s) stochastic result yields all possible realizations and hence contains the full spectrum of uncertainties. Thus it will enable the analyst to obtain a more complete description of the resulting load ensemble at large, calculate its statistics, and eventually arrive at more precise estimates of e.g. the probability of exceedance of some load threshold more quickly.

Recently, progress has been made towards stochastic analysis of wind turbines. For example, results have been shown for an aeroelastic analysis with one uncertain system parameter, stiffness or damping (Desai and Sarkar, 2010), and for the a stochastic formulation of airfoil lift, drag, and pitching moment in stall conditions (Bertagnolio et al., 2010). Moreover, stochastic models have been used for wake modeling, treating wake center and shape as random processes (Doubrawa et al., 2016). However, only very early steps have been completed to include the biggest source of uncertainty: the uncertain inflow from turbulent atmospheric wind. On a wind farm scale Padrón et al. (2016) recently presented a layout optimization based on a polynomial chaos formulation for the freestream wind speed and direction. Finally Guo (2013) offers a stochastic wind model used for a stochastic analysis of wind turbine loads. However, he still bases the stochastic analysis on deterministic sampling (i.e. path B in Fig. 1). Moreover the stochastic wind model this model driven by the decomposition (bi-orthogonal and Karhunen-Loève) of a specific set of wind field data. It hence is not generally applicable, but relies on the availability of sufficient data.

As turbulent wind is already represented as a stochastic field in many common wind models, a transition from a deterministic aerodynamic model for specific wind realizations, to a stochastic model yielding the whole stochastic load ensemble at once, seems an obvious step. However, this step comes with a simple, yet fundamental challenge: current wind models, even simple spectral models, rely on a large number of random variables to set the wind sample's phase angles. Since realizations of large sets of random variables can be generated very quickly, this is not a problem for deterministic load analyses. However, the computational cost of stochastic analysis methods increases dramatically with the number of random variables included, a fact commonly known as the 'curse of dimensionality' (Majda and Branicki, 2012). This renders current wind models inaccessible to stochastic methods, and thus poses a major barrier to the further development of stochastic models for the analysis of wind



turbine loads based on a stochastic description of the turbulent wind input.

To address this problem we reformulate an industry standard wind model into a reduced order engineering model. The aim of our work is to develop a wind model that can generate a realistic wind field with appropriate (long term) dynamic properties

from considerably less random variables than the current models. In the last three decades numerous turbulent wind models have been proposed. Kleinhans et al. (2008) summarizes of a few. However, none of the previous models had an application in stochastic aerodynamic models in mind. Hence, as generating a large set of random numbers as a seed for a wind field realization is usually no problem, the existing models we are aware of rely on a large number of random variables – too large to be applicable to a direct stochastic modeling of the aerodynamic wind turbine equations (path C in Fig. 1).

In the following we focuses on a formulation for the IEC standard spectral wind description (IEC 61400-1, Ed. 3, 2005), so that it may be directly useful in industry. (Veers, 1988) was chosen as baseline and starting point. This model is widely used, for example in the stochastic wind simulator *TurbSim* (described by Jonkman and Kilcher (2012)), which synthesizes a sample of turbulent atmospheric wind from Veers' spectral formulation. Although it is well known that Veers' model does not

capture all physical details of 'real' atmospheric wind (e.g. Mücke et al. (2011); Morales et al. (2012); Lavely et al. (2012); Park et al. (2015)), it is for many cases an appropriate engineering model (Nielsen et al., 2007). Due to its comparatively high independence of site specific parameters, ease of use, and low resource requirements, Veers' model is the preferred model for many applications (Lavely et al., 2012). Moreover, it is endorsed by the governing wind turbine design standard IEC 61400-1, Ed. 3 (2005), and thus is widely used in the wind energy industry. This underlines that its fidelity is accepted as a reasonable

compromise in engineering practice for wind energy. As such, Veers' model provided a well accepted foundation to base further development on. Note that our goal is not improving on known deficiencies of Veers' model, but to arrive at a model that can generate a wind samples of comparable (and accepted) fidelity with significantly less random variables, geared towards eventual inclusion in a stochastic wind turbine simulation.

The following sections will first briefly review Veers' model to set the stage for the proposed modifications. Subsequently, the new reduced order wind model is introduced, and finally results are presented, which confirm that key statistical properties (cross-correlation, covariance, auto- and cross-spectrum) are conserved by the new model. The paper concludes by giving direction for continued work on integrating the wind model into a turbine simulation and on refinements with other turbulent wind descriptions. To not overload this paper, the focus is solely on the details and validation of the stochastic wind inflow

model itself. Interested readers should refer to Fluck and Crawford (2016a) for the basic stochastic aerodynamic model, or Fluck and Crawford (2016c) for an example how the reduced order wind model is used to calculate stochastic loads on a stationary wing.



## 2 Method

In this section we first briefly summarize Veers' method, as it represents the established method for synthesizing turbulent wind (Nielsen et al., 2007; Lavely et al., 2012), and at the same time is the baseline for our contribution. Subsequently, we will introduce our new reduced order method. Note that section 2.1 is only meant as a summary to lay out the basics for the

following work. For a complete introduction, the reader is referred to Veers' original paper (Veers, 1988) and successive work, e.g. (Kelley, 1992; Nielsen et al., 2004; Burton et al., 2011).

### 2.1 Veers' method

In a spectral method, the wind speed time series $u_k(t)$ at each point $P_k$, $k = 1...N_P$, in the sampled wind field is obtained through the inverse discrete Fourier transform of a set of discrete frequencies components $U_{mk}$ at $\omega_m = 2\pi f_m$, $m = 1...N_F$

$$u_k(t) = \sum_m U_{mk} e^{i\omega_m t} \tag{1}$$

Here $m$ is used to index the frequency bins, and $k$ is used to index the points in space where wind speed data is recorded. Usually, the terms $U_{mk}$ are binned Fourier amplitudes centered at the frequency $\omega_m$, prescribed by the wind speed spectrum $S(\omega_m)$ at each point $P_k$. Often a Kaimal spectrum is used (IEC 61400-1, Ed. 3, 2005).

Following Veers' method (Veers, 1988), $U_{mk} \in \mathbb{C}$ contains not only the amplitude but also the random phase angles at point $P_k$ for each frequency $\omega_m$. To obtain the desired coherence for all frequencies and between any two points in the wind field, all phase angles

$$\theta_{mk} = \arctan\left(\frac{\text{Im}(U_{mk})}{\text{Re}(U_{mk})}\right) \tag{2}$$

need to be correlated correctly. To achieve this, Veers multiplies the set of $N_R = N_F \cdot N_P$ independent, uniformly distributed
random variables $\xi_{jm} \sim U(0,1)$ with the weighting tensor $H_{jkm}$, obtained from the discrete cross spectrum $S_{jk}(\omega_m)$, to obtain the complex Fourier coefficients $U_{mk}$ at each frequency band $\omega_m$:

$$U_{mk} = \sum_{j=1}^{k} H_{jkm} e^{i\,2\pi\xi_{jm}} \tag{3}$$

where $S_{jk}(\omega_m)$ is given by the relevant design standard or physics model. Note that through Eq. 3 the phase angles at point $P_k$ are related to the phases at all previously computed points $P_{j<k}$. Thus, correctly correlated Fourier coefficients are obtained,
which can now be inserted into Eq. 1 to obtain a correlated wind field.

  This method works well to generate multiple (deterministic) data sets at many points. However, as already noted by Veers in his original publication (Veers, 1988), Eq. 3 changes the amplitude of each Fourier coefficient, such that $|U_{mk}| \neq \sqrt{S_{kk}(\omega_m)}$ for all but the point computed first. Thus, the prescribed (e.g. Kaimal) spectrum $S_{kk}(\omega_m)$ is not conserved anymore at each




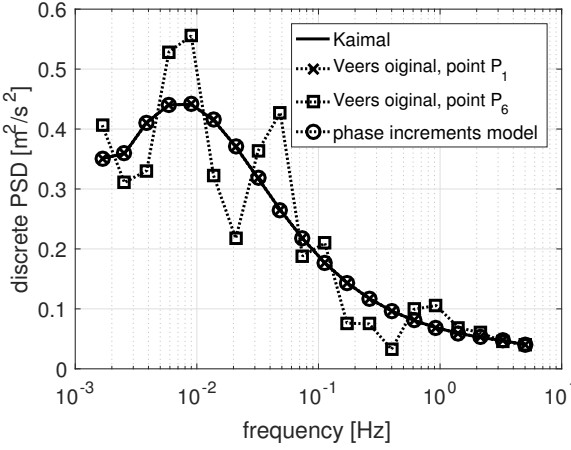

**Figure 2.** Raw wind spectra from a single wind speed sample, no averaging. Kaimal: the analytic spectrum; Veers: sample of the spectrum resulting from Eq. 3 at two different points $P_1$ and $P_6$; Phase increments: the spectrum from the reduced order phase increment model Eq. 8 (identical for all points).

point for any single realization, see Fig. 2. However, if spectra are averaged over either several points or several realizations, the wind field's average spectrum converges to the prescribed spectrum as $\lim_{N \to \infty} 1/N \sum_{k=1}^{N} |U_{mk}(\omega_m)| = \sqrt{S_{kk}(\omega_m)}$, with $N$ the number of samples or realizations. This means the field still is stochastically homogeneous, as expected. However, for a stochastic analysis where only a limited number of samples might be used, this may pose a challenge. In the following we

introduce a reduced order model based on phase angle increments. This model not only yields a significant reduction in random variables required to synthesize a stochastic wind field, but it also analytically preserves the prescribed spectrum at any single point for each realization (see Fig. 2 'phase increments').

## 2.2 The reduced order model with phase increments

To arrive at a reduced order model we follow a two step process. First is a reduction in the number of frequencies necessary for the spectral composition of the wind speed time series at a single point in space, and with it a reduction in the number of random phase angles associated with each frequency. This frequency reduction has been done before. For example Fluck and Crawford (2016a) showed that with ten frequencies from the IEC Kaimal spectrum, logarithmically spaced in $[0.003, 5]$ Hz (a $T = 333$ s sample, resolved at 10 Hz, a reasonable time step for wind turbine simulations, cf. Bergami and Gaunaa (2014)),

a realistic wind speed time series can be produced, with probability distribution (and thus turbulence intensity), as well as the wind speed auto-correlation similar to results from a full *TurbSim* simulation at 10 Hz for 10,000 Hz (roughly $5 \cdot 10^4$ frequency bins). Ten frequencies, and thus ten random variables for the phase angles is manageable as input to a stochastic model. However, when dealing with a wind field big enough to be used for wind turbine calculations, many points (typically a grid in the



order of $15 \times 15$ points over the rotor disk) of correlated wind speed are necessary. The challenge is to extend this limited frequency wind description from a single point to a spatially varying wind field without excessively increasing the number of random variables required. Fung et al. (1992) introduced a wind model which models both the spatial and the temporal dimension through Fourier modes. They reduced the number of modes down to as little as 38, however, the model then relied on several random numbers associated with each mode. Fung et al. (1992) did not report in detail how many random variables they used for their model, but the equations indicated that this number was still considerably larger than manageable by stochastic methods. The following paragraphs will introduce a new approach, which will allow to create a stochastic wind field from a significantly reduced number of random variables, independently of the (spatial) size of the wind field, i.e. independently of both, the number of data points over the rotor disc as well as the lateral extent.

In Veers' model the phase angle matrix $\mathbf{\Theta} = [\theta_{mk}]$ is populated with random numbers. We note that random phase angles in the rows and columns of $\mathbf{\Theta}$ carry out two distinctly different functions. At each individual point $P_i$ the different phase angles in the column vector $[\theta_m]_i = \mathbf{\Theta}_i$ generate constructive/ destructive interference of the ensemble of base sinusoids. Thus, different realizations of $\mathbf{\Theta}_i$ generate the "gusty" nature of the wind speed time series at that point. This is indeed the *temporal* variability of the wind.[1] On the other hand, the wind speed structure in space, for example the fact that strong winds at one point correlate with strong winds at a nearby point, is captured through the relation of phase angles for the one particular frequency $\omega_l$ at different points $P_i$ and $P_j$ – that is in each row of $\mathbf{\Theta}$, $[\theta_k]_l = \bar{\mathbf{\Theta}}_l$. This is the *spatial* variability of the wind.

While the phase angles at each point (the columns $\mathbf{\Theta}_i$) are uncorrelated, the phase angles between two points (the rows $\bar{\mathbf{\Theta}}_l$) have to be correlated to reproduce the spatial structure correctly. For two column vectors $[\theta_m]_i$ and $[\theta_m]_j$ this means while the entries within each vector are uncorrelated the two vectors themselves are element-wise correlated, Fig. 3. For wind, this correlation decreases with both, increasing frequency and increasing distance.

For our use-case of turbulent wind as input to dynamic wind turbine analysis, we observe the following:

1. The temporal variability is of primary importance, since it drives the dynamic excitation of the system under investigation. This is the duration of gusts and lulls, captured by the energy distribution in the frequency spectrum of the wind sample.

2. The spatial variability needs to be represented correctly to yield representative wind loads, since the length scale of spatial wind structures needs to be correct to result in the correct integral loads. For example, for any instance when a sensor *A* on the blade experience an increased load, another sensor *B* a certain distance away from *A* needs to experience a load correctly correlated to the load at *A*.

3. For each point, all elements in each column vector $\mathbf{\Theta}_i$ are independent (Fig. 3). However, the column vectors $\mathbf{\Theta}_i$ and $\mathbf{\Theta}_j$ at two points $P_i$ and $P_j$ are element-wise correlated. This means the phases in each row vector $\bar{\mathbf{\Theta}}_l$ are *not* independent.

---

[1]Note that one dimension of the block of wind, in the average wind direction, corresponds to a temporal correlation as the wind moves downwind.



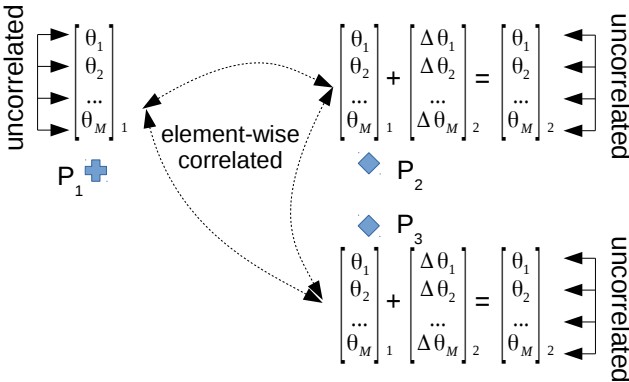

**Figure 3.** Schematic of random phase angle vectors and deterministic phase increments.

Following Veers' method, only the elements in $\boldsymbol{\Theta}_1$ are independent, while the phases at all other points are mapped from i.i.d. random variables $\xi_{mi}$ such that they are correlated to the phases at the base point $P_1$ (and thus to each other), Eq. 3.

To obtain a reduced order model which requires fewer random variables, i.e. fewer phase angles, we propose splitting the

5 complex Fourier coefficients $U_{mk}$, into a temporal and a spatial part. The temporal part will contain the amplitude of each Fourier mode as well as the random phase angles. It therefore will determine the structure of the wind speed sample in time. The spatial part will contain the phase correlation between different points across the wind field. It will thus set the wind field structure in space. To reflect this approach we can write:

$$U_{mk} = \underbrace{U_{m1}}_{\text{temporal}} \cdot \underbrace{e^{i\Delta\theta_{mk}}}_{\text{spatial}} \qquad (4)$$

The temporal part contains the amplitude according to the prescribed power spectrum $S(\omega_m)$ and a vector of random phase angles $\theta_{m1} = 2\pi\xi_m$ at an arbitrary base point $P_1$ within the wind field:

$$U_{m1} = \sqrt{S(\omega_m)}\, e^{i\theta_{m1}} \qquad (5)$$

with independent and identically distributed $\xi_m \sim U(0,1)$ as before. Similar to the velocity increments used for wind speed

15 interpolation by Fluck and Crawford (2016b), the spatial part is based on the idea of phase increments $\Delta\theta_{mk}$, which are specific to each point and each frequency relative to the base point $P_1$:

$$\Delta\theta_{mk} = \theta_{mk} - \theta_{m1} \qquad (6)$$

The increment $\Delta\theta_{mk}$ holds the correlated phase information to generate the correct spatial structures. Since $\theta_{mk}$ and $\theta_{m1}$ are random numbers, the increments $\Delta\theta_{mk}$ are random, too. In contrast to Veers' approach of employing the cross spectrum to



map a set of uncorrelated random variables to a set of correlated phases for each point in the wind field, we neglect the random nature of $\Delta\theta_{mk}$ and consider the phase increments deterministic constants to move between points as illustrated in Fig. 3. Note that $\Delta\theta_{mk}$ only contains the spatial structure, but not the temporal part. That means 'gusty' features of the wind (lulls and gusts at different points) are still generated from random numbers; only the wind field's structure in space is fixed with each

specific set of phase increments. Based on the three observations above (1-3) this seems justified for two reasons. Firstly, the phases in each row vector $\bar{\Theta}_l$ are correlated, while the phases in each column vector $\Theta_i$ are uncorrelated (3). This means there is more 'randomness' in the temporal dimension then in the spatial dimension. Secondly, for the dynamic analysis of a wind energy device, the temporal part is of primary importance. While the spatial structures have to be represented correctly, their variability can be considered secondary (1,2).

Nonetheless, it is important to note that focusing on the temporal part does not mean that each realization of the reduced order wind field will exhibit the same spatial structure of gusts and lulls, i.e. that a gust at point $P_i$ would e.g. necessarily come with a lull at another point $P_j$. Instead, gusts and lulls result from the interference of different frequency component sinusoids and phase offsets. Based on the specific realization $\theta_{m1}$ the phase angles at each point $\theta_{mk} = \Delta\theta_{mk} + \theta_{m1}$ will be

different each time. Thus, the interference between the frequency components and consequently the structure of the gusts and lulls will be different with each different realization of phases at the base point $\theta_{m1}$. Figure 6, which will be discussed later, demonstrates this fact.

Inserting Eqs. 5 and 6 into Eq. 4 yields the Fourier coefficients based on only one vector of random phases $\theta_{m1}$ and the

(auto-) spectrum:

$$U_{mk} = \sqrt{S(\omega_m)}\,e^{i(\theta_{m1}+\Delta\theta_{mk})} \tag{7}$$

Substituting $\theta_{m1} = 2\pi\xi_m$ with $\xi_m \sim U(0,1)$ as before, Eq. 1 can be turned into our reduced order model:

$$u_k(t) = \sum_m \sqrt{S(\omega_m)}\,e^{i(\omega_m t + 2\pi\xi_m + \Delta\theta_{mk})} \tag{8}$$

Note that while Eq. 3 changes the amplitude of each Fourier coefficient and thus distorts the spectrum at each point, Eq. 8 fully

conserves the spectrum.

In contrast to Veers' original model, where $N_R = N_F \cdot N_P$, in the reformulated model $N_R = N_F$. This means the number of random variables $N_R$ only depends on the number of frequencies $N_F$ used for the wind Fourier series, not on the number of wind speed measurement points $N_P$ in the 3D wind field. With the available strategies to reduce the number of frequencies

required in a spectral wind model Eq. 8 now allows expression of a turbulent wind field consistent with Veers' model, but with significantly fewer frequencies.

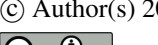



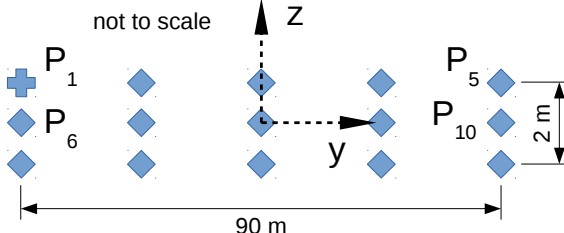

**Figure 4.** Schematic of grid points of wind speed data (minimal test case). We arbitrarily chose the right hand top point ($P_1$) to be the base point.

What remains is to obtain the phase angle increments $\Delta\theta_{mk}$. Since these determine the cross-correlation between any two points in the wind field, and since the cross-correlation and the cross-spectrum are linked as a Fourier transform pair (correlation theorem, see e.g. (Kauppinen and Partanen, 2011)), it should be possible to analytically generate one set (one realization) of phase increments directly from the cross-spectrum. For now, however, we extract one phase angle increment set from one
realization of Veers' Eq. 3, with an analytic solution left to future work.

## 3   Results and discussion

In the following we will take a closer look at statistical metrics of the synthetic reduced order wind field. As mentioned earlier our goal is not to develop a more physically faithful wind model, but rather to reduce the number of random variables required
while retaining similar fidelity as the methods currently in use. *TurbSim* (Jonkman and Kilcher, 2012) is widely used in industry and the de facto standard to generate synthetic wind fields for wind turbine analysis. Hence we use *TurbSim* wind speed data sets as the benchmark. In the following we compare results obtained from *TurbSim* to two different reduced order models.

The first is our implementation of Veers' model, which allowed us to freely choose the number of frequencies at each data
point and the frequency binning. As suggested by Veers equations implementation relies on the conventional inverse discrete Fourier transform with random phase angles at each frequency bin. This model was validated directly against *TurbSim*. If many frequencies are used and identical phase angles are enforced perfect agreement of the resulting data set was found as expected. As shown by Fluck and Crawford (2016a) the wind speed time series at a single point for a 333 s sample can be well represented with $N_f = 10$ logarithmically spaced frequencies $\boldsymbol{f} = \boldsymbol{\omega}/(2\pi) = [f_m] \in [f_1, f_{N_F}] = [0.003, 5]$ Hz with $f_m = 10^{a_k}$
and $a_m = \log_{10}\left(\frac{f_{N_F}}{f_1}\right)\frac{m-1}{N_F-1}$ for $m = 1, \ldots, N_F$. For better comparison (equal sample length $T = 600$ s) we use $N_F = 20$ frequencies in $\boldsymbol{f} = [f_m] \in [1/600, 5]$ Hz. I; the results of this model are labeled '**Veers**$_{\text{red}}$' in the following discussion. This model does not include new theory, yet it is a critical step between *TurbSim* (and thus Veers' original model) and our reduced order model. The second model presented is our reduced order model as described above (Eq. 8). The newly introduced theory of deterministic phase increments $\Delta\theta$ is employed here, together with a limited number frequencies $N_F = 20$ and thus a





reduced number of stochastic variables $N_R = 20$. These results are labeled '**Veers** $_{\text{red}, \Delta\theta}$'.

**Table 1.** Comparison of random numbers used in different wind models for a common grid size.

|  | *TurbSim* | Veers $_{\text{red}}$ | Veers $_{\text{red}, \Delta\theta}$ |
| --- | --- | --- | --- |
| sample length | 10 min | 10 min | 10 min |
| grid size $N_{Py} \times N_{Pz}$ | $15 \times 15$ | $15 \times 15$ | $15 \times 15$ |
| frequencies $N_F$ | $\sim$3,000 | 20 | 20 |
| total number of random variables $N_R$ | $6.75 \cdot 10^5$ | 4,500 | 20 |

Tab. 1 gives a comparison of the three models. Note particularly the total number of random variables required by each model, assuming a typical grid resolution in the order of $15 \times 15$ points over a rotor disk of $D = 90$ m diameter. While the

use of a limited set of frequencies (Veers $_{\text{red}}$) yields a noticeable reduction in random numbers, for a turbulent wind field with several wind speed data points in x- and y-direction, this alone is not enough to arrive at a wind model with few enough random numbers to be applicable in a stochastic method (several dozen random variables to be tractable). Only the additional introduction of deterministic phase increments (Veers $_{\text{red}, \Delta\theta}$) to decouple the number of random variables from the number of wind speed data points reduces the number of random variables drastically enough to obtain a wind model which can be reasonably

handled by a stochastic method.

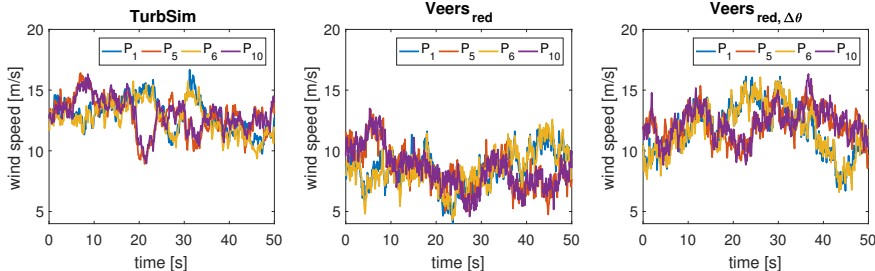

**Figure 5.** Three 50 s excerpt of a wind speed time series sample at four points generated from different models and different random seeds. TurbSim: NREL's original *TurbSim* model; Veers $_{\text{red}}$: Veers model with a limited number of frequencies ($N_F = 20$); Veers $_{\text{red}, \Delta\theta}$: Veers model with a limited number of frequencies and phase increments to model spatial structures at different points. See Tab.

As a test case we study a wind field generated on $N_{Py} \times N_{Pz} = 5 \times 3 = 15$ points located on a regular grid as depicted in Fig. 4. The origin of the wind field was located 100 m above ground with mean wind speed $\bar{u} = 10$ m/s and no wind shear. The IEC class A normal turbulence model with a Kaimal spectrum (IEC 61400-1, Ed. 3, 2005) and homogeneous turbulence

was used. Data was sampled at 10 Hz. We arbitrarily chose the top left hand point ($P_1$) as the base point. Note that the grid used here contains fewer points than the usual grid for the analysis of a modern $D = 90$ m rotor diameter wind turbine (where





$N_{Py} = N_{Pz} = 15$ is more likely). However, the reduced number of grid points enabled us to solve the equations quickly with all models and more clearly illustrate the method. At the same time, the configuration of Fig. 4 still allowed us to study both the wind speed time series of points in close proximity (e.g. $P_1$ and $P_6$), as well as at more distant points (e.g. $P_1$ and $P_5$).

Fig. 5 shows realizations of the wind speed time series sampled at four points ($P_1$, $P_5$, $P_6$, and $P_{10}$ in Fig. 4) from the three different models. For each model the samples are generated from different random seeds. Thus the time series are not identical. Still, it can be seen that the fundamental structures are conserved through both reduced order models. In particular, even if wind samples are synthesized with only 20 random numbers and deterministic phase increments (Veers $_{\text{red}, \Delta\theta}$) the wind speeds at two points in close proximity ($P_1$ and $P_6$, or $P_5$ and $P_{10}$) are highly correlated, while at more distant points (e.g. $P_1$ and $P_5$)

the correlation is weaker. It is important to note that this holds not only for points in relation to the base point, but for all point pairs. For example, points $P_5$ and $P_{10}$ are both far away from the base point, but close to each other. As expected, the wind speeds at these two points are well correlated.

Fig. 6 shows three realizations of wind speed time series plots at three points obtained from the new phase increment model

(Veers $_{\text{red}, \Delta\theta}$), Eq. 8. The phase increments are considered deterministic, and $\Delta\theta_{mk}$ is fixed for all realizations. The randomness enters the time series only via random phase angles at the base point $P_1$ with $\theta_{m1} = 2\pi\xi_m$. As can be seen from the figure, this does *not* result in a complete determination of the spatial relation between wind speeds at different points, since the samples still contain different gusts and lulls at different instances in time.

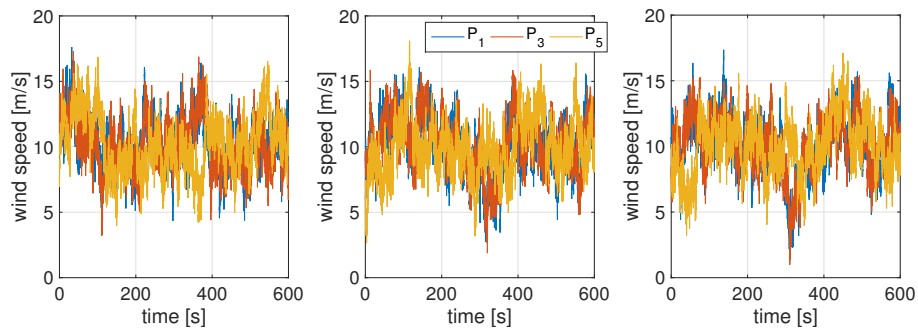

**Figure 6.** Three realizations of wind speed time series at three points generated from the the new reduced order model with fixed phase increments (Veers $_{\text{red}, \Delta\theta}$ model).

Beyond this qualitative visual comparison of the wind speed time series the remainder of this section will show that the phase increment model produces the same statistics as Veers' original model (with only 20 frequencies) as well as the full *TurbSim* model (with the full set of frequencies) for the most important statistical metrics.



## 3.1 Cross-correlation

Fig. 7 compares the cross-correlation for two different point pairs, $P_1$-$P_5$ (90 m apart) and $P_1$-$P_6$ (1 m apart) as obtained from six 99 s windows from a 600 s sample from our reduced model with fixed phase increments (Veers $_{\mathrm{red}, \Delta\theta}$), from Veers' model with 20 frequencies (Veers $_{\mathrm{red}}$), and from the full *TurbSim* simulation. To reduce noise and compare meaningful (rather than

possibly extreme) values the results are presented as averages of 100 realizations from different random seeds for both phase angles and phase increments.

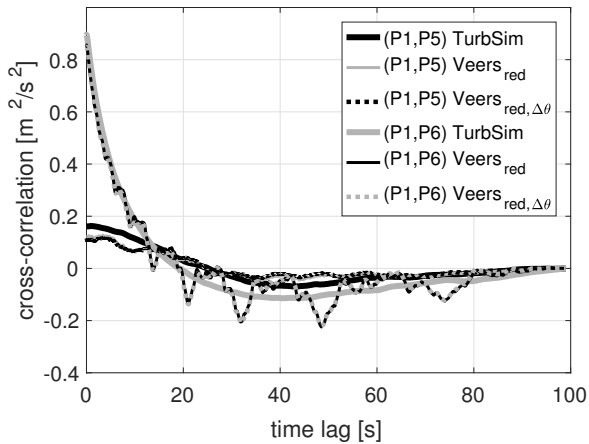

**Figure 7.** Wind speed cross-correlation for two point pairs generated from different models; a close pair ($P_1$,$P_6$) and a distant one ($P_1$,$P_5$).

As can be seen from the figure the cross-correlation in general agrees very well for both the close points (gray lines) and the distant point pair (black lines). The results from our implementation of Veers' model and from the phase increment model

are almost identical and hence difficult to distinguish in Fig. 7. Note that the *TurbSim* data is smoother, presumably due to the significantly higher number of frequencies contained in the *TurbSim* data set.

Further investigation with the pair $P_5$ and $P_{10}$, two points close to each other but far away from the base point $P_1$ (not included in Fig. 7), shows that for all three models the cross-correlation is almost identical to the curve for $P_1$-$P_6$. This

confirms that with our phase increment model the cross-correlation of the homogeneous turbulence field, and with it the length scale of spatial structures, is indeed only dependent on the distance between two points, but not on the two specific points themselves.

## 3.2 Covariance

Now we look at the covariance as a function of the distance between two points and compare data from *TurbSim* to the 20 fre-

quency of Veers' model (Veers $_{\mathrm{red}}$) and to our reduced model with phase increments (Veers $_{\mathrm{red}, \Delta\theta}$). As above we use averages



from 99 s windows out of 100 realizations of 600 s samples.

From Fig. 8 it can be seen that our implementation of Veers' model agrees well with the results from *TurbSim*. The phase
increment model, however, yields slightly, but consistently less covariance. A more detailed investigation reveals the reason
for this: the covariance depends on the cross-spectrum and thus the spectrum at each individual point. Consequently the
discrepancy between the covariance functions is connected to the fact that Veers' model distorts the spectrum at each individual
point, such that with Eq. 3 $|U_{mk}| = \sqrt{\tilde{S}_{mk}} \neq \sqrt{S_{mk}}$ (see discussion in section 2.1). When we replace $S$ by the distorted
spectrum $\tilde{S}$ at each particular point $P_k$ in Eq. 8 all three curves do match. However, $\tilde{S}$ does not in fact represent the prescribed
Kaimal spectrum. Thus we conclude that our phase increment model actually represents the desired covariance better than
Veers' original model and *TurbSim*.

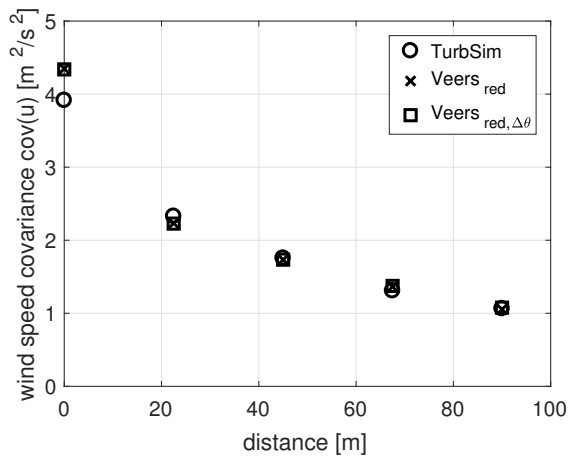

**Figure 8.** Wind speed covariance for points different distances apart.

### 3.3 Cross-spectrum

Next we compare the cross-spectrum, which is again obtained as the average spectrum from 100 realizations (from different
random seeds). However, this time 6,000 s were sampled to obtain sufficiently long data sets for a proper resolution of the
low frequency components. Note that the same set of 20 frequencies $[f_k] \in [1/600, 5]$ Hz are used for both the 20 frequency
(Veers $_{red}$) and the phase increment (Veers $_{red, \Delta\theta}$) implementations. Hence the $T = 6,000$ s signal repeats after 600 s. The
spectrum is binned into discrete bins of frequencies $f_m$ equal to the logarithmically spaced frequencies initially used to generate
the wind speed time series. Fig. 9 shows a comparison of the cross-spectra estimates for different point pairs obtained through
Welch's periodogram method employed on the full 6,000 s samples with no extra windowing (Welch, 1967). We study the base
point, and its closest neighbor ($P_1$-$P_6$); the base point and a point far away ($P_1$-$P_5$); and a point pair close together, but far away
from base point ($P_5$-$P_{10}$). For reference, the prescribed Kaimal spectrum $S$ is included, as well as the analytic cross-spectrum





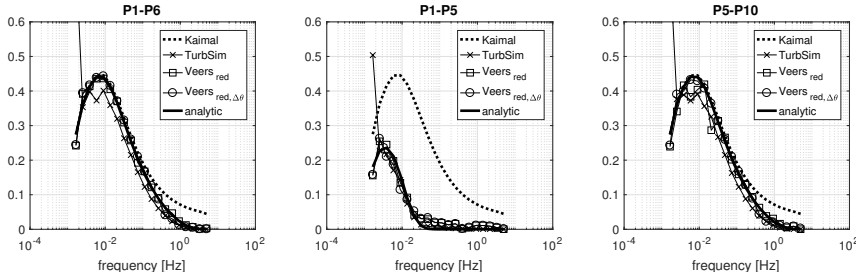

**Figure 9.** Wind speed cross power spectral density for three point pairs from different models, together with the analytic results (Eq. 9) and the prescribed Kaimal auto-spectrum. Left: the base point, and its closest neighbor. Middle: the base point and a point far away. And right: a point pair close together, but far away from base point.

obtained by:

$$S_{ij} = Coh\sqrt{S_{ii}S_{jj}} \tag{9}$$

from the (auto-) spectra $S_{ii} = S$ and the coherence function $Coh$ as defined by the standard IEC 61400-1, Ed. 3 (2005).

Again, the phase increment model (Veers $_{\text{red}, \Delta\theta}$) in all cases reproduces the analytic spectrum well with only 20 random variables. This time, however, the *TurbSim* results do not match as well. The reason is that *TurbSim* chooses the lowest frequency $f_1$ and the frequency bin width $\Delta f$ such that $\Delta f = f_1 = 1/T$, and thus uses a wider frequency band for the first bin compared to our logarithmically spaced bins. When re-binning to the logarithmic range this results in excess power (and an artificial peak) in the first bin and hence less power in higher frequency bins. Note, however, that this is an artifact of the
discrete spectrum and the frequency binning, and not a discrepancy in the underlying data.

### 3.4   Outlook: wind turbine rotor blade loads

To further assess the validity of the reduced order wind model, loads were calculated for one single blade on a three bladed $R = 35$ m diameter wind turbine rotor spinning at a tip speed ratio $\lambda = 6.1$. Loads were obtained at $\Delta t = 0.1$ s time steps
through a simple blade element momentum model supplied with wind generated either from *TurbSim*, or from our reduced model with fixed phase increments (Veers $_{\text{red}, \Delta\theta}$) on a $15 \times 15$ grid of data points over the rotor disc. The hub height is set to $h_{hub} = 90$ m, with the hub height mean wind speed $\bar{u} = 12$ m/s, power law wind shear with power law exponent $a = 0.2$ (according to Jonkman and Kilcher (2012)), and IEC normal turbulence model, class A (IEC 61400-1, Ed. 3, 2005). Fig. 10 shows the probability distribution $p(T)$ of thrust loads $T$ on one blade calculated from 100 realizations of a 600 s wind field.
*TurbSim* used the full set of roughly 3,000 frequencies at each of the $15 \times 15$ grid points. The reduced order model, on the other hand, relied on only 20 frequencies with all 100 realizations generated one set of fixed phase increments. It can be seen, that





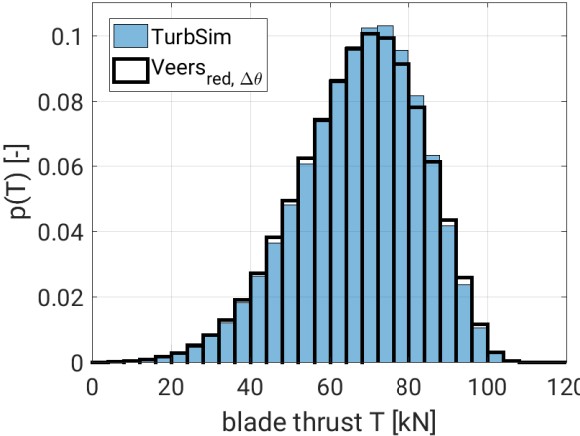

**Figure 10.** Blade thrust load probability distribution from BEM model based on wind fields generated with either *TurbSim* or from the reduced order Veers model with constant phase increments.

the reduced order model, although relying on significantly fewer random variables ($N_R = 6.75 \cdot 10^5$ versus $N_R = 20$ for each realization, see Tab. 1), produces almost the same load probability distribution.

## 3.5  Discussion

As shown by the results presented in this section the phase increments wind model presented in section 2.2 can reproduce important statistics (both of a wind field, as well as for resulting wind loads) with the same accuracy as the full model. At the same time, the phase increments model requires significantly less random variables. As indicated by Fig. 6 the phase increments model does *not* produce identical spatial structures with each realization, even thought a large part of the spatial randomness is neglected is Eq. 8. This further illustrates the method's ability to retain important stochastic information.

The results from sections 3.1-3.3 are generated from a set of 100 different phase increments generated from 100 different random seeds. This was necessary because, due to the random equations, it was not possible to compare the results from a single realization. This might have resulted in uncharacteristically bad (or good) agreement only by the chance of comparing 'bad' (or 'good') realizations. Instead only the averages over multiple realizations could be compared. For a stochastic analysis as outlined in the introduction, however, only a very limited set of phase increment realizations would be used. Hence, some part of the randomness of the wind field will be lost. This is the price to be payed for using a reduced order model. In section 2.2 we justify this choice. The results, particularly Figs. 6 and 10, support the notion that a very limited set of phase increment realizations, or even a single one, can be sufficient. It is still to be determined, however, how many sets will actually be necessary for adequate results, and how the associated reduction in randomness influences the relevant output quantities, e.g. for a wind turbine analysis the resulting loads, especially the probability of extreme loads. Preliminary results for wind





turbine blade loads calculated from a Blade Element Momentum model indicate that only one single set of phase increments is sufficient to obtain almost the same statistical load distribution as from the conventional analysis based on standard *TurbSim* wind fields.

## 4   Conclusions

5   Stochastic analysis and uncertainty quantification are generally very active fields of research in engineering with the developed methods increasingly adopted by industry. To enable practitioners to apply these methods to wind turbine aerodynamics and more generally wind loading analysis on various structures, we presented a new method, which significantly reduces the number of random variables used in the wind model. This reduction is critical, because the computational effort of the common stochastic solutions is very sensitive to the number of random variables involved.

The model introduced here employs a separation of the temporal (correlation in time) and spatial (coherence in space) part of the random dimension of turbulent wind. While the temporal part is still determined from random variables, the spatial part is collapsed into deterministic phase increments. Thus the number of random variables is reduced by several orders of magnitude compared to the commonly used model developed by Veers and implemented in *TurbSim*, currently the (de facto) standard tool

15   for synthetic wind generation. A comparison of the most important stochastic metrics (cross-correlation, covariance, auto- and cross-spectrum) showed that the reduced order model based on phase increments still reproduces these metrics as accurately as Veers' equations or *TurbSim*. Moreover, preliminary results were presented, which indicate that the reduced order wind model based on phase increments also preserves wind turbine blade loads well. A detailed study quantifying the impact of using deterministic phase increments on the overall statistics of wind turbine loads is yet to be carried out. Subsequent to the

20   implementation of this reduced order wind model in a full wind turbine simulator, which is the focus of ongoing work, these ultimate questions can be addressed.

*Acknowledgements.*   We gratefully acknowledge the funding provided for this study by the Pacific Institute for Climate Solutions (PICS), the German Academic Exchange Service (DAAD), and the Natural Sciences and Engineering Research Council of Canada (NSERC).



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
