# Peer review of "An engineering model for 3D turbulent wind inflow based on a limited set of random variables"

_Wind Energy Science, 2017_

## Referee Comment (RC1) · Anonymous Referee #1 · 7 Apr 2017

Review of the manuscript "An engineering model for 3D turbulent wind inflow based on a limited set of random variable" by Fluck and Crawford.

General comments:

This manuscript proposes a method of decoupling temporal randomness from the spatial variability of a turbulent wind speed field. The spatial variability of wind speed is modelled deterministically, whereas the temporal fluctuations are addressed through the traditional approach of the random phase angles. The authors compared their results two other methods (TurbSim software and modified Veers method).

The manuscript is well written and the illustrations are nice. I believe the manuscript is well suited for Wind Energy Science, but few clarifications need to be addressed before the manuscript can be published in this journal. My comments are separated in three sections: (1) moderate (the most important) comments, (2) minor comments and (3) comments related to the grammar and style of writing. My moderate comments are focused around (1) the underlying hypothesis that temporal variability in wind dynamics is more important than the spatial variability and (2) the applicability of the proposed method to highly transient (in both space and time) wind events.

Therefore, I assign a moderate revisions for this manuscript. Although I have a long list of comments, I believe most of them can be implemented in relatively short amount of time.

Moderate comments:

1. P8 & 9. What are the references and studies that would confirm your statement that the temporal variability is more important than the spatial variability of the wind in a dynamic wind analysis? You need to provide some proofs for your observations 1-3. Reading your observations 1 and 2, I conclude that both are of the same importance, but on Page 10 (L5-10) you claim that temporal variability is more important based on the observations 1 and 2.

2. P9 & 10. Regarding Eq. (3), you established that $\Delta\theta_{mk}$ is random variable (P9, L19), but later you decided to neglect the fact it is a random variable (P10, L2). What is a justification that a random process can be replaced by a deterministic constants? I guess this comment boils down to my previous comment.

3. P9 and later in the text. The concept of $\Delta\theta_{mk}$ means that the correlation between two points is space is always the same for a given frequency and those two points. For example, at a frequency bin centered around 10 Hz, and for the points P1 and P2, $\Delta\theta_{mk}$ is always the same number. Similarly, for a frequency bin around, say, 0.1 Hz and the points P1 and P2, $\Delta\theta_{mk}$ is once again always the same value (but not the same as the one for 10 Hz). If I am correct, please provide an example like this that would simplify the concept to the reader (not necessarily this one; I am sure you can formulate it better).

4. P10. L9. Is there a study that showed the temporal variability is more important than the variability of the spatial structure of wind field?

5. Figure 7. Why is there a large difference between your model and TurbSim model for the times lags less than 10 s? Why does your model give higher cross-correlations for small time lags than TurbSin for time lags less than approximately 10 s, but the cross-correlations are relatively similar for time lags larger than about 20 s? Please discuss this observation in text.

6. Your Figure 10 is a very valuable part of your paper. You should emphasize more on this figure. For example, it is one of the answers for my major comments above. However, I believe you still need to provide the strong proof that your assumption is valid in a more general case.

7. Lastly, what do you think how valid your method would be for the case of highly unsteady winds (e.g. gust fronts, downbursts, etc.). I would like to see a critical dissection on this topic in Section 3.5. You can start with the following references:

Chay MT, Albermani F, Wilson R. 2006. Numerical and analytical simulation of downburst wind loads. *Engineering Structures* **28**(2): 240–254. DOI: 10.1016/j.engstruct.2005.07.007.

Chen L, Letchford CW. 2004. A deterministic–stochastic hybrid model of downbursts and its impact on a cantilevered structure. Engineering Structures 26(5): 619–629. DOI: 10.1016/j.engstruct.2003.12.009.

Solari G, De Gaetano P, Repetto MP. 2015. Thunderstorm response spectrum: Fundamentals and case study. Journal of Wind Engineering and Industrial Aerodynamics 143: 62–77. DOI: 10.1016/j.jweia.2015.04.009.

Solari G. 2016. Thunderstorm response spectrum technique: Theory and applications. *Engineering Structures* **108**: 28–46. DOI: 10.1016/j.engstruct.2015.11.012.

See the references in those papers for more literature on the subject of highly space-time varying thunderstorm winds. I believe that you should test your method for one of these wind phenomena in your future work.

Minor comments:

1. P4, L24. What is that you want to say in the sentence "Moreover the stochastic wind…" Could not understand it due to the incorrect English.

2. P5, L7-9. This sentence is very difficult to understand. For example, "the existing models we are aware of rely…" What message you want to convey?

3. P5, L12. It should be Veers (1988). Please pay attention how you cite references when it comes to their in-line appearance in text.

4. P6. L2-6. For the most part, you don't need this small paragraph here. Most of this information is given in the last paragraph on Page 5 and the last sentence with the references can be moved to Section 2.1. Please modify accordingly.

5. P6. L13. First, you should have a comma after "Often". Second, the sentence sounds incomplete. Try, for example, "In many cases, Kaimal spectrum is used to represent the wind time series in the frequency domain [ref]". Also, add the reference for the Kaimal spectrum:

J.C. Kaimal, J.C. Wyngaard, Y. Izumi, O.R. Cote. 1972. Spectral characteristics of surface-layer turbulence. *Quarterly Journal of the Royal Meteorological Society*, 98, pp. 563–589.

6. Figure 2. If I am interpreting this figure correctly, the Kaimal, Veers origina (Point P1) and phase increment model all collapse to the same line. Please clarify additionally the caption for this figure.

7. P8, L15. In the footnote, what do you mean by "the block of wind"?

8. P11, L21. What do you mean by "I; the results" and why is Veers$_{red}$ in bold?

9. P12, L1. Similar to my previous comment, is there a particular reason to have Veers$_{red,\Delta\Theta}$ in bold? You are not writing it in bold afterwards.

10. Figure 5 caption. It should read "See Tab. 1 for additional information." at the end of the caption.

11. Figure 6. Only the plot in the middle has the legend, which is not consistent with Figure 5. Moreover, please move the legend within the frame of the plot, as it is in Figure 5.

12. Eq. 9. Coherence function is a function and thus it should not be italicized (e.g., sin, cos, ln, etc. are not italicized). Please write Coh without being italicized in Eq. 9 and the line following Eq. 9.

13. Increase the size of Figure 9 and therefore the fonts. Please indicate on the plots that the y-axis are cross power spectral densities.

14. Figure 10 caption. State that BEM stands for Blade Element Theory.

15. References. Sometimes the titles of journal articles have capitalized each word (first letter) and sometimes you used the sentence style (i.e., only the first word capitalized). Please be consistent.

Grammatical and stylistic comments:

1. P2, L3. I believe you should not have a space after "/" in "and/or"

2. P2, L6 & 7: You need a comma after "In a numerical experiment, …". In the same sentence, is it "base" or "based"?

3. P2, L 14 & 15. Two consecutive sentences start with "This is …" Please reword it to increase readability.

4. P3, L1. How about "In wind turbine engineering, the driving force is the turbulent atmospheric wind which is commonly described …" and then you don't need the comma after "field"

5. P4, L11. Put a comma after "Instead".

6. P4, L17, You have "tha a stochastic…" Please correct.

7. P4, L21 & 22. Put a comma after "On a wind farm scale, ". Also, have a comma after "Finally, "

8. P4, L25. It should be "Hence, it is not generally…"

9. P4, L27-29. After "wind models, a transition from …" you don't need the other commas in that sentence.

10. P5, L6. Summarizes a few or summarizes of a few? I think the former is more accurate.

11. P5, L11. It should read "In the following, we focus on…" or even better "In this study, we focus on…"

12. P7, L4. Instead of "in the following", maybe it sounds better if you use "here,".

13. P8, L13. I believe you don't need a space after "/" (Similar to my first comment).

14. P11, L8 & L13. Put a comma after "In the following,."

15. P11, L15. Put a comma after "As suggested by Veers equations,."

16. P11, L20. Put a comma after the closed bracket (i.e., before "we").

17. P12, L11. Put a comma after "As a test case,."

18. P15, L8. Put a comma after "Eq. 8,."

19. P15, L12. Put a comma after "Next,."

---

## Referee Comment (RC2) · Anonymous Referee #2 · 14 Jun 2017

The authors presented an interesting study in particular in revealing new results that have not been published earlier. The manuscript is well designed in structure and clearly presented. To this reviewer, the manuscript is worth being published. However, there are few minor issues need to be further clarified or revised. Below, the issues are presented.

1. Page 5, line 15-17: Within these lines it is stated that Veer's turbulent wind model is chosen as baseline. However, as it is mentioned, it does not capture all physical details of real atmospheric wind. It would be informative to include the main advantage of this model compared to other models, which makes it an appropriate model for many cases.

2. Page 7, line 10-15: How the reduction in the number of frequencies is performed? That is based on what physical criterion?

3. Page 8, line 1: The grid is set to be 15 x 15 points over the rotor disc. Is there any experimental of theoretical justification for this number of points?

4. Page 11, line 19: The number of frequencies is set to be 10. What is the reason to choose this number? Does the change in the number affect the results? If not, it seems one can reduce the number of frequencies as low as possible.

5. Page 12, line 12: It would be more appropriate to replace figure 4 such that it is closer to the place where it is mentioned within the manuscript.

6. Page 12: It would be more informative to present figure 5 in larger size.

7. Page 12, line 16: It is stated that "the grid used here contains fewer points than usual grid for the analysis of a modern D=90 m rotor diameter wind turbine." What is the appropriate number of grid points for the abovementioned turbine (need to provide appropriate citation)? What is the justification for this decision in the study? Does the difference affect the results?

8. Page 15, line 2-10: based on what is stated within these lines, the figure shows good agreement of Veers' model with the results from Turbsim. However, the figure does not confirm the claim. In contrast, it presents that the Veers' model and increment model have better agreement in terms of trend and magnitude in comparison with the Turbsim results. At the end of the paragraph, it is claimed that the phase increment model represents the desire covariance. However, this conclusion is not clearly justified in particular based on what is presented in the figure.

9. Page 16, line 7-8: Is there any experimental or theoretical justification for choosing logarithmically spaced bin?

10. Page 18, line 15: It is stated that the comparison of the stochastic metrics was used to evaluate the model. Among the metrics, auto spectrum is also mentioned which is not presented within the manuscript.

---

## Author Comment (AC1) · 27 Jun 2017

**An engineering model for 3D turbulent wind inflow based on a limited set of random variables**

Fluck and Crawford

**Response to comments from anonymous reviewer #1**

Moderate comments:

1.  *P8 & 9. Confirming that the temporal variability is more important than the spatial variability.*
    This is an original thought derived from Veers' model. We have not come across any literature studying this fact.
    Here is another way to look at it. Hopefully this clarifies our thoughts:
    Assuming Veer's spectral model is adequate, we note that phase angles are independent over frequencies $Cov(\Theta_{mk}, \Theta_{nk})=0$, but somehow dependent between points $|Cov(\Theta_{mk}, \Theta_{ml})|>0$. Without further discussion of the actual structure of this dependence we infer that the phase angle matrix **Θ** contains more randomness in the columns (time) than in the rows (space). Thus we decide to include randomness in time, but neglect randomness in space. The results presented in section 3 confirm the validity of this assumption.

    We will revise the manuscript to better reflect these thoughts.

2.  *P9 & 10. Regarding Eq. (3), choosing $\Delta\Theta_{mk}$ to be deterministic.*
    This indeed boils down the comment 1:
    In order to arrive at a reduced order model something needs to be simplified. We choose to set $\Delta\Theta_{m(kl)}= \Theta_{ml} - \Theta_{mk}$ as deterministic constant. Thus we generate a realization for one part of the stochastic process, i.e. we collapse a part of the random domain of the problem into one sample, but we retain randomness in the other part. Comment 1 provides the reason for this choice, and although this means we neglect the randomness in the rows of **Θ**, and with it some part of the randomness in the wind field, the good agreement of our results with Veers original wind model justify this choice.

3.  *P9 and later in the text. The concept of $\Delta\Theta_{mk}$.*
    This is a nice way to put paraphrase our concept. We will include something along these lines to more graphically explain that concept to the reader.

4.  *P10. L9. Spatial and temporal variability.*
    See comment 1. This is a new thought, and we have not come across any studies in this direction.

5.  *Figure 7. Difference between for the times lags less than 10 s?*
    The cross correlation curves do not perfectly agree. The difference is most prominent in the oscillations of the (P1,P5) curve, and in the (P1-P6) curves around 10s and 50s. We do not think that this is related to high or low frequencies, but due to the fact that our model uses significantly less frequencies than TurbSim. Hence the turbulent energy is discretized over fewer modes in our model, and the curves are not as smooth.

6. *More emphasis on Figure 10.*
   We fully agree. However, representing the underlying BEM model and the modifications made to transform it into a stochastic BEM model would exceed the scope of a single paper. These results will be presented in a separate paper (Fluck and Crawford: "A fast stochastic solution method for the Blade Element Momentum equations for long-term load assessment", Wind Energy, submitted). A reference will be included in the next revison.

7. *Validity of our method would be for the case of highly unsteady winds.*
   Our method (like Veers' method) is a spectral representation of a random process. The wind field is generated through inverse Fourier transform with random phase increments. Thus the underlying process is (per definition) assumed to be stationary.
   Consequently, fronts, downbursts, and other singular transient events can naturally not be modelled directly in any spectral model.

   It might be possible to model these event through superposition of a discrete deterministic event with our random model, similar to Chay et al. (2006) who superposes a deterministic downburst profile over an ARMA model for the turbulent fluctuations. However, it is our goal to derive a formulation for wind as input to stochastic models. These models will have difficulties dealing with singular deterministic events themselves. We doubt if moving this way is beneficial. Nonetheless, this could obviously be the subject of a future study.

Minor comments:

1. *P4, L24. Correction of "Moreover the stochastic wind…"*
   This should be: "Moreover this model is driven by the decomposition (bi-orthogonal and Karhunen-Loève) of a specific set of wind."

2. *P5, L7-9 difficult to understand.*
   We change this to: "However, none of the previous models had an application in stochastic aerodynamic models in mind. Since random numbers can be generated very quickly, existing models rely on a large set of random variables to be used as a seed for a wind field realization. However, this random seed usually contains too many random variables to be applicable to a direct stochastic modeling of the aerodynamic wind turbine equations (path C in Fig. 1)."

3. *P5, L12. It should be Veers (1988).*
   Thanks for pointing this out. We have changed citation style before submission and obviously missed adopting references to the new style at a few places.

4. P6. L2-6.
   We believe it is important to state why Veers model is briefly revisited. To avoid repetition, we rephrase this paragraph:
   "Veers' method represents the established method for synthesizing turbulent wind (Nielsen et al., 2007; Lavely et al., 2012) and at the same time is the baseline for our contribution. Hence the method is briefly summarized here to lay out the basics for the following work. For a complete introduction the reader is referred to Veers' original paper (Veers, 1988) and successive work, e.g. Kelley (1992); Nielsen et al. (2004); Burton et al. (2011)."

5. *P6. L13.*
   We will reword this accordingly and include the suggested reference.

6. *Figure 2.*
   We will clarify this in the figure.

7. *P8, L15, "block of wind".*
   We will discard the footnote and reword the sentence: "This is indeed the streamwise and thus the temporal variability of the wind field."

8. *P11, L21. "l; the results" and* **Veers**$_{red}$*.*
   This is a typo, which we will remove. We will also remove the bold typeface for the labels.

9. *9. P12, L1.*
   We will also remove the bold typeface for the labels.

10. *Figure 5 caption.*
    We will change this accordingly.

11. *Figure 6 caption.*
    We will change this accordingly.

12. *Eq. 9.*
    We will change this accordingly.

13. *Figure 9.*
    We will change this accordingly.

14. *Figure 10 caption.*
    We will change this accordingly.

15. *References.*
    We will double check this.

Grammatical and stylistic comments:

We will revise the manuscript and correct the errors pointed out.

---

## Author Response (AR1)

**An engineering model for 3D turbulent wind inflow based on a limited set of random variables**

Fluck and Crawford

**Response to comments from anonymous reviewer #1**

Moderate comments:

1. *P8 & 9. Confirming that the temporal variability is more important than the spatial variability.*
   This is an original thought derived from Veers' model. We have not come across any literature studying this fact.
   Here is another way to look at it. Hopefully this clarifies our thoughts:
   Assuming Veer's spectral model is adequate, we note that phase angles are independent over frequencies $Cov(\Theta_{mk}, \Theta_{nk})=0$, but somehow dependent between points $|Cov(\Theta_{mk}, \Theta_{ml})|>0$. Without further discussion of the actual structure of this dependence we infer that the phase angle matrix $\Theta$ contains more randomness in the columns (time) than in the rows (space). Thus we decide to include randomness in time, but neglect randomness in space. The results presented in section 3 confirm the validity of this assumption.
   We have revised the manuscript (see pp. 8, 9) to better reflect these thoughts.

2. *P9 & 10. Regarding Eq. (3), choosing $\Delta\Theta_{mk}$ to be deterministic.*
   This indeed boils down the comment 1:
   In order to arrive at a reduced order model something needs to be simplified. We choose to set $\Delta\Theta_{m(kl)}= \Theta_{ml} - \Theta_{mk}$ as deterministic constant. Thus we generate a realization for one part of the stochastic process, i.e. we collapse a part of the random domain of the problem into one sample, but we retain randomness in the other part. Comment 1 provides the reason for this choice, and although this means we neglect the randomness in the rows of $\Theta$, and with it some part of the randomness in the wind field, the good agreement of our results with Veers original wind model justify this choice.
   We have revised the manuscript (see p. 10) to better reflect these thoughts.

3. *P9 and later in the text. The concept of $\Delta\Theta_{mk}$.*
   We have adopted this nice way to put paraphrase our concept. See changes under Eq. (6), p. 10. We hope this will explain our concept more graphically.

4. *P10. L9. Spatial and temporal variability.*
   See comment 1. This is a new thought, and we have not come across any studies in this direction.

5. *Figure 7. Difference between for the times lags less than 10 s?*
   The cross correlation curves do not perfectly agree. The difference is most prominent in the oscillations of the (P1, P5) curve, and in the (P1, P6) curves around 10s and 50s. We do not think that this is related to high or low frequencies, but due to the fact that our model uses significantly less frequencies than *TurbSim*. Hence the turbulent energy is discretized over fewer modes in our model, and the curves are not as smooth.

6. *More emphasis on Figure 10.*
   We fully agree. However, representing the underlying BEM model and the modifications made to transform it into a stochastic BEM model would exceed the scope of a single paper. These results will be presented in a separate paper (Fluck and Crawford: "A fast stochastic solution method for the Blade Element Momentum equations for long-term load assessment", Wind Energy, submitted). We included a reference at the end of Section 3.4.

7. *Validity of our method would be for the case of highly unsteady winds.*
   Our method (like Veers' method) is a spectral representation of a random process. The wind field is generated through inverse Fourier transform with random phase increments. Thus the underlying process is (per definition) assumed to be stationary.
   Consequently, fronts, downbursts, and other singular transient events can naturally not be modelled directly in any spectral model.

   It might be possible to model these event through superposition of a discrete deterministic event with our random model, similar to Chay et al. (2006) who superposes a deterministic downburst profile over an ARMA model for the turbulent fluctuations. However, it is our goal to derive a formulation for wind as input to stochastic models. These models will have difficulties dealing with singular deterministic events themselves. We doubt if moving this way is beneficial. Nonetheless, this could obviously be the subject of a future study.

   We have included this in Section 3.5.

Minor comments:

1. *P4, L24. Correction of "Moreover the stochastic wind…"*
   This should be: "Moreover this model is driven by the decomposition (bi-orthogonal and Karhunen-Loève) of a specific set of wind."
   We have changed this accordingly.

2. *P5, L7-9 difficult to understand.*
   We have changed this to: "However, none of the previous models had an application in stochastic aerodynamic models in mind. Since random numbers can be generated very quickly, existing models rely on a large set of random variables to be used as a seed for a wind field realization. However, this random seed usually contains too many random variables to be applicable to a direct stochastic modeling of the aerodynamic wind turbine equations (path C in Fig. 1)."

3. *P5, L12. It should be Veers (1988).*
   We have reviewed our references and believe they are correct now.

4. P6. L2-6.
   We believe it is important to state why Veers model is briefly revisited. To avoid repetition, we rephrase this paragraph:
   "Veers' method represents the established method for synthesizing turbulent wind (Nielsen et al., 2007; Lavely et al., 2012) and at the same time is the baseline for our contribution. Hence

the method is briefly summarized here to lay out the basics for the following work. For a complete introduction the reader is referred to Veers' original paper (Veers, 1988) and successive work, e.g. Kelley (1992); Nielsen et al. (2004); Burton et al. (2011)."

5. *P6. L13.*
   We have reworded this accordingly.

6. *Figure 2.*
   We have clarified this in the caption.

7. *P8, L15, "block of wind".*
   We have discarded the footnote and reword the sentence: "This is indeed the streamwise and thus the temporal variability of the wind field."

8. *P11, L21. "I; the results" and **Veers**$_{red}$.*
   This is a typo, which was removed. We also removed the bold typeface for the labels.

9. *9. P12, L1.*
   We have also removed the bold typeface for this label.

10. *Figure 5 caption.*
    We have changed this accordingly.

11. *Figure 6 caption.*
    We have changed this accordingly.

12. *Eq. 9.*
    We have changed this accordingly.

13. *Figure 9.*
    We have changed this accordingly.

14. *Figure 10 caption.*
    We have changed this accordingly.

15. *References.*
    We have changed this accordingly.

Grammatical and stylistic comments:

We have revised the manuscript and correct the errors pointed out.

**Response to comments from anonymous reviewer #2**

1. *Page 5, line 15-17: main advantage of Veers' model.*
   We consider three main advantages of Veers model and write:
   "Due to its comparatively high independence of site specific parameters, ease of use, and low resource requirements, Veers' model is the preferred model for many applications (Lavely et al., 2012)."

2. *Page 7, line 10-15: reduction of numbers of frequencies.*
   In a frequency model the frequency bins as chosen arbitrarily, but such that the energy in the frequency spectrum can be represented adequately. As discussed on P5, L7-9, using a large set of frequencies (i.e. a large set of random numbers) was not a problem for deterministic models. However, moving to stochastic models requires a significant reduction of random variables, and hence frequencies. We show, that using 10 to 20 random variables for 10 to 20 frequencies is sufficient. This choice is not driven by physical arguments, but by the limitations of the stochastic models where such a reduced order wind field formulation is to be used.
   We have amended the first paragraph of Section 2.2. to clarify this.

3. *Page 8, line 1: experimental of theoretical justification for grid resolution.*
   The choice of grid resolution is a tradeoff between accuracy and computational effort. For a typical D = 90 m diameter rotor 15 x 15 points is a common choice. We change the bracket to "(for a D = 90 m diameter rotor somewhere in the order of 15 x15 points over the rotor disk is are typically used)".

4. *Page 11, line 19: The number of frequencies.*
   The number of frequencies does affect the quality of the resultant wind field. Too few frequencies result either in periodic wind speeds or distinct wind speed oscillations. Eight to ten frequencies for a 333 s sample was found to be the minimum.
   We hope the additions to the first paragraph of Section 2.2. clarify this, too.

5. *Location of Figure 4.*
   We leave this to the final setting in the journal print.

6. *Size of Figure 5.*
   We have increased the size.

7. *Page 12, line 16: grid resolution.*
   In this study we are not concerned with wind turbine loads. Hence it is not necessary to model the wind field over the full wind turbine rotor disc. To assess the reduced order

wind model it is important to show that it adequately represents wind field properties both for close and distant points. In order to do this without unnecessary computational effort, we use a study grid, which contains only a few close and a few distant points. To make this clearer we changed p13, ll 6ff to:

"Since the goal here is not to calculate wind turbine loads, but to merely asses the quality of the reduced order wind model, we used a dummy wind field generated on $N_{Py}$ x $N_{Pz}$ = 5 x 3 = 15 points located on a regular grid as depicted in Fig. 4. This is fewer points than the usual grid for the analysis of a modern D = 90 m rotor diameter wind turbine. However, the reduced number of grid points enabled us to solve the equations quickly with all models and more clearly illustrate the method. At the same time, the configuration of Fig. 4 still allowed us to study both the wind speed time series of points in close proximity (e.g. P1 and P6), as well as at more distant points (e.g. P1 and P5). The origin [...]"

8. *Page 15, line 2-10: model names*
   Thank you for catching this! We erroneously swapped model names in line 3. We changed this paragraph (p. 13, ll 2ff):

"From Fig. 8 it can be seen that the covariance from all three model agrees fairly well. Our implementations of Veers' model, Veers$_{red}$ and Veers$_{red, \Delta\Theta}$, which both use a limited set of frequencies, agree almost perfectly. The TurbSim version with the full set of roughly 3,000 frequencies, on the other hand, yields slightly different covariance. A more detailed investigation reveals the reason for this: the covariance depends on the cross-spectrum and thus the spectrum at each individual point. Consequently, the discrepancy between the covariance functions is connected to the fact that Veers' model distorts the spectrum at each individual point, such that with Eq. 3 $|U_{mk}| = \sqrt{\tilde{S}_{mk}} \neq \sqrt{S_{mk}}$ (see discussion in section 2.1). When we replace S in our implementation by the distorted spectrum $\tilde{S}$ at each particular point $P_k$ in Eq. 8 all three curves do match. However, $\tilde{S}$ does not in fact represent the prescribed Kaimal spectrum. Thus we conclude that our phase increment model actually represents the desired covariance better than Veers' original model and *TurbSim*."

9. *Page 16, line 7-8: logarithmically spaced bins.*
   With a logarithmic spacing more bins are located at low frequencies and fewer at high frequencies. This enables us to represent the wind and its spectrum more efficiently. For clarity we changed on p. 12, ll. 13-14 to:

"[...] with $N_f$ = 10 logarithmically spaced frequencies, which allowed a more efficient representation of the wind and its spectrum. We set f= [...]"

10. *Page 18, line 15: auto spectrum*

> The auto-spectrum is included in Fig. 2. We realize that this might get lost towards the end of the paper. We have changed the heading of section 3.3. to "Power spectra" and changed the beginning of the section:

"Wind speed spectra are again obtained as average from 100 realizations (from 100 different random seeds). However, this time 6,000 s were sampled to obtain sufficiently long data sets for a proper resolution of the low frequency components. Note that the same set of 20 frequencies $[f_k] \in [1 = 600;\ 5]$ Hz are used for both the 20 frequency (Veers$_{red}$) and the phase increment (Veers$_{red,\ \Delta\Theta}$) implementations. Hence the T = 6,000 s signal repeats after 600 s. The spectrum is binned into discrete bins of frequencies $f_m$ equal to the logarithmically spaced frequencies initially used to generate the wind speed time series.

The wind speed auto-spectrum is included in Fig. 2. By definition (Eq. 8) the reduced order model produces the prescribed (auto-) spectrum exactly. 
[revised manuscript text omitted]

20 need to be correlated correctly. To achieve this, Veers  starts with a set of $N_R = N_F \cdot N_P$ independent, uniformly distributed random variables $\xi_{jm} \sim U(0,1)$  and multiplies these with a weighting tensor $H_{jkm}$, obtained from the discrete cross spectrum $S_{jk}(\omega_m)$ (given by the relevant design standard or physics model), to obtain the complex Fourier coefficients $U_{mk}$  for each frequency band $\omega_m$:

$$U_{mk} = \sum_{j=1}^{k} H_{jkm} e^{i\, 2\pi \xi_{jm}} \tag{3}$$

25  Through Eq. 3 the phase angles at each point $P_k$ are related to the phases at all previously computed points $P_{j<k}$. Thus, correctly correlated Fourier coefficients are obtained, which can now be inserted into Eq. 1 to obtain a correlated wind field.

This method works well to generate multiple (deterministic) wind speed data sets at many points. However, as already

[revised manuscript text omitted]

While the phase angles at each point (the columns $\boldsymbol{\Theta}_i$) are uncorrelated, the phase angles between two points (the rows $\bar{\boldsymbol{\Theta}}_l$) have to be  somehow dependent on each other to reproduce the spatial structure correctly. For two column vectors $[\theta_m]_i$ and $[\theta_m]_j$ this means while the entries within each column vector are uncorrelated , $(cov(\theta_{mk}, \theta_{nk}) = 0)$ the elements within each vectors are not independent, i.e. correlated $(|cov(\theta_{mk}, \theta_{ml})| > 0)$, cf. Fig. 3. For wind, this correlation decreases with both, increasing frequency and increasing distance.

 Based on these observation, we note the following for our use-case of turbulent wind as input to dynamic wind turbine analysis :
* * *
[1]

[Figure]

**Figure 3.** Schematic of random phase angle vectors and deterministic phase increments.

1. The temporal variability (in the columns of $\Theta$) is of primary importance, since it drives the dynamic excitation of the system under investigation. This is the  structure of gusts and lulls, captured by the energy distribution in the frequency spectrum of the wind sample.

2.  Also the spatial variability (in the rows of $\Theta$) needs to be represented correctly to yield representative wind loads,  which eventually result in the correct integral loads. For example,  at any instance when a sensor *A* somewhere on the blade experience an increased load, another sensor *B* a certain distance away from *A* needs to experience a load correctly correlated to the load at *A*. However, since there will necessarily be some averaging of the loads across the blades this is of secondary importance.

3. For each point, all elements in each column vector $\boldsymbol{\Theta}_i$ are independent (Fig. 3). However, the column vectors $\boldsymbol{\Theta}_i$ and $\boldsymbol{\Theta}_j$ at two points $P_i$ and $P_j$ are element-wise correlated. This means the phases in each row vector $\bar{\boldsymbol{\Theta}}_l$ are *not* independent. Following Veers' method, only the elements in $\boldsymbol{\Theta}_1$ are independent, while the phases at all other points are mapped from i.i.d. random variables $\xi_{mi}$ such that they are correlated to the phases at the base point $P_1$ (and thus to each other), Eq. 3. This means there is more "randomness" in the columns of $\Theta$ than in the rows – an important fact, which we will soon exploit.

To obtain a reduced order model which requires fewer random variables we propose splitting the complex Fourier coefficients $U_{mk}$, into a temporal and a spatial part. The temporal part will contain the amplitude of each Fourier mode as well as the random phase angles. It therefore will determine the structure of the wind speed sample in time. The spatial part will contain the phase correlation between different points across the wind field. It will thus set the wind field structure in space. To reflect this

approach we can write:

$$U_{mk} = \underbrace{U_{m1}}_{\text{temporal}} \cdot \underbrace{e^{i\Delta\theta_{mk}}}_{\text{spatial}} \tag{4}$$

The temporal part contains the amplitude according to the prescribed power spectrum $S(\omega_m)$ and a vector of random phase angles $\theta_{m1} = 2\pi\xi_m$ at an arbitrary base point $P_1$ within the wind field:

$$U_{m1} = \sqrt{S(\omega_m)}\, e^{i\theta_{m1}} \tag{5}$$

with independent and identically distributed $\xi_m \sim U(0,1)$ as before. Similar to the wind speed increments used for wind interpolation by Fluck and Crawford (2016a), the spatial part is based on the idea of phase increments $\Delta\theta_{mk}$, which are specific to each point and each frequency relative to the base point $P_1$:

$$\Delta\theta_{mk} = \theta_{mk} - \theta_{m1} \tag{6}$$

The increment $\Delta\theta_{mk}$ holds the correlated phase information to generate the correct spatial structures. Since $\theta_{mk}$ and $\theta_{m1}$ are random numbers, the increments $\Delta\theta_{mk}$  should be random, too.  However, in contrast to Veers' approach of employing the cross spectrum to map a set of uncorrelated random variables to a set of correlated phases for each point in the wind field, we neglect the random nature of $\Delta\theta_{mk}$ and consider the phase increments deterministic constants This means that for different wind field realizations the correlation coefficient between two points in space is fixed for each frequency (n.b. this is the stochastic correlation; It does *not* establish a deterministic one-to-one dependence of the wind speeds at two points!). For example, for a frequency bin at $f_1 = 10$ Hz and point $P_4$, $\Delta\theta_{14}$ has the same value for each realization. Similarly, for a frequency bin $f_5 = 0.1$  Hz and the same point, $\Delta\theta_{54}$ has once again always the same value (but but different from $\Delta\theta_{14}$).

Assuming $\Delta\theta_{mk}$ to be constant is the core assumption of the presented reduced order model. It clearly is a simplification, but essential to arriving at a model reliant on a reduced number of random variables. The results presented in Section 3 will confirm the validity of this assumption. Note moreover that $\Delta\theta_{mk}$ only contains the spatial structure, but not the temporal part. That means 'gusty' features of the wind (lulls and gusts at different points) are still generated from random numbers; only the wind field's  spatial correlation is fixed with each specific set of phase increments. Based on the three observations above (1-3) this seems justified for two reasons. Firstly, the phases in each row vector $\bar{\boldsymbol{\Theta}}_l$ are correlated, while the phases in each column vector $\boldsymbol{\Theta}_i$ are uncorrelated (3). This means there is more 'randomness' in the temporal dimension then in the spatial dimension. Secondly, for the dynamic analysis of a wind energy device, the temporal part is of primary importance. While the spatial structures have to be represented correctly, their exact variability can be considered secondary (1,2).

It is important to note that focusing on the temporal part does not mean that each realization of the reduced order wind field will exhibit the same spatial structure of gusts and lulls, i.e. that a gust at point $P_i$ would e.g. necessarily come with a

lull at another point $P_j$.  In contrary, the proposed method does not alter the original correlation between wind speeds at two  points which is generally smaller than unity. Graphically speaking gusts and lulls result from the interference of different frequency component sinusoids and phase offsets. Based on the specific realization $\theta_{m1}$ the phase angles at each point $\theta_{mk} = \Delta\theta_{mk} + \theta_{m1}$ will be different each time. Thus, the interference between the frequency components and consequently the

5 structure of the gusts and lulls will be different with each different realization of phases at the base point $\theta_{m1}$. Figure 6, which will be discussed later, demonstrates this fact. Nonetheless, the proposed reduction of random variables necessarily causes a certain increase in dependence of wind speeds across the wind field. An investigation into how this dependence actually looks like in detail  will be the subject of future work.

10 Inserting Eqs. 5 and 6 into Eq. 4 yields the Fourier coefficients based on only one vector $[\theta_{m1}]$ of random phases  and the (auto-) spectrum:

$$U_{mk} = \sqrt{S(\omega_m)}\, e^{i(\theta_{m1} + \Delta\theta_{mk})} \tag{7}$$

Substituting $\theta_{m1} = 2\pi\xi_m$ , Eq. 1 can be turned into our reduced order model (with $\xi_m \sim U(0,1)$ as before):

[revised manuscript text omitted]

 Since the goal here is not to calculate wind turbine loads, but to merely asses the quality of the reduced order wind model, we used a dummy wind field generated on $N_{Py} \times N_{Pz} = 5 \times 3 = 15$ points located on a regular grid as depicted in Fig. 4.  This is fewer points than the usual grid for the analysis of a modern  D = 90 m rotor diameter wind turbine . However, the reduced number of grid points enabled us to solve the

equations quickly with all models and more clearly illustrate the method. At the same time, the configuration of Fig. 4 still allowed us to study both the wind speed time series of points in close proximity (e.g. $P_1$ and $P_6$), as well as at more distant points (e.g. $P_1$ and $P_5$). The origin of the wind field was located 100 m above ground with mean wind speed $\bar{u} = 10$ m/s and no wind shear. The IEC class A normal turbulence model with a Kaimal spectrum and homogeneous turbulence was used (IEC 61400-1, Ed. 3, 2005). Data was sampled at 10 Hz. We arbitrarily chose the top left hand point ($P_1$) as the base point.

Fig. 5 shows realizations of the wind speed time series sampled at four points ($P_1$, $P_5$, $P_6$, and $P_{10}$ in Fig. 4) from the three different models. For each model the samples are generated from different random seeds. Thus the time series are not identical. Still, it can be seen that the fundamental structures are conserved through both reduced order models. In particular, even if wind samples are synthesized with only 20 random numbers and deterministic phase increments (Veers$_{\text{red},\,\Delta\theta}$) the wind speeds at two points in close proximity ($P_1$ and $P_6$, or $P_5$ and $P_{10}$) are highly correlated, while at more distant points (e.g. $P_1$ and $P_5$) the correlation is weaker. It is important to note that this holds not only for points in relation to the base point, but for all point pairs. For example, points $P_5$ and $P_{10}$ are both far away from the base point, but close to each other. As expected, the wind speeds at these two points are well correlated.

Fig. 6 shows three realizations of wind speed time series plots at three points obtained from the new phase increment model (Veers$_{\text{red},\,\Delta\theta}$), Eq. 8. The phase increments are considered deterministic, and $\Delta\theta_{mk}$ is fixed for all realizations. The randomness enters the time series only via random phase angles at the base point $P_1$ with $\theta_{m1} = 2\pi\xi_m$. As can be seen from the figure, this does *not* result in a complete determination of the spatial relation between wind speeds at different points, since the samples still contain different gusts and lulls at different instances in time.

[Figure]

**Figure 6.** Three realizations of wind speed time series at three points generated from the the new reduced order model with fixed phase increments (Veers$_{\text{red},\,\Delta\theta}$ ).

Beyond this qualitative visual comparison of the wind speed time series the remainder of this section will show that the phase increment model produces the same statistics as Veers' original model (with only 20 frequencies) as well as the full *TurbSim* model (with the full set of frequencies) for the most important  metrics.

**3.1 Cross-correlation**

Fig. 7 compares the cross-correlation for two different point pairs, $P_1$-$P_5$ (90 m apart) and $P_1$-$P_6$ (1 m apart) as obtained from six 99 s windows from a 600 s sample from our reduced model with fixed phase increments (Veers $_{red, \Delta\theta}$), from Veers' model with 20 frequencies (Veers $_{red}$), and from the full *TurbSim* simulation. To reduce noise and compare meaningful (rather than possibly extreme) values the results are presented as averages of 100 realizations from different random seeds for both phase angles and phase increments.

[Figure]

**Figure 7.** Wind speed cross-correlation for two point pairs generated from different models; a close pair ($P_1$,$P_6$) and a distant one ($P_1$,$P_5$).

As can be seen from  Fig. 7 the cross-correlation in general agrees very well for both the close points and the distant point pair. The results from our implementation of Veers' model and from the phase increment model are almost identical and hence difficult to distinguish Note that the *TurbSim* data is smoother, presumably due to the significantly higher number of frequencies contained in the *TurbSim* data set.

Further investigation with the pair $P_5$ and $P_{10}$, two points close to each other but far away from the base point $P_1$ (not included in Fig. 7), shows that for all three models the cross-correlation is almost identical to the curve for $P_1$-$P_6$. This confirms that with our phase increment model the cross-correlation of the homogeneous turbulence field, and with it the length scale of spatial structures, is indeed only dependent on the distance between two points, but not on the two specific points themselves.

**3.2 Covariance**

Now we look at the covariance as a function of the distance between two points and compare data from *TurbSim* to the 20 frequency of Veers' model (Veers $_{red}$) and to our reduced model with phase increments (Veers $_{red, \Delta\theta}$). As above we use averages

from 99 s windows out of 100 realizations of 600 s samples.

From Fig. 8 it can be seen that  the covariance from all three model agrees fairly well. Our implementations of Veers' model, Veers $_{\text{red}}$ and Veers $_{\text{red, }\Delta\theta}$, which both use a limited set of frequencies, agree almost perfectly. The *TurbSim* version with the full set of roughly 3,000 frequencies, on the other hand, yields slightly different covariance. A more detailed investigation reveals the reason for this: the covariance depends on the cross-spectrum and thus the spectrum at each individual point. Consequently, the discrepancy between the covariance functions is connected to the fact that Veers' model distorts the spectrum at each individual point, such that with Eq. 3 $|U_{mk}| = \sqrt{\tilde{S}_{mk}} \neq \sqrt{S_{mk}}$ (see discussion in section 2.1). When we replace $S$ in our implementation by the distorted spectrum $\tilde{S}$ at each particular point $P_k$ in Eq. 8 all three curves do match. However, $\tilde{S}$ does not in fact represent the prescribed Kaimal spectrum. Thus we conclude that our phase increment model actually represents the desired covariance better than Veers' original model and  *TurbSim*.

[Figure]

**Figure 8.** Wind speed covariance for points different distances apart.

**3.3 Power spectra**

 Wind speed power spectra are again obtained as  average from 100 realizations (from 100 different random seeds). However, this time 6,000 s were sampled to obtain sufficiently long data sets for a proper resolution of the low frequency components. Note that the same set of 20 frequencies $[f_k] \in [1/600, 5]$ Hz are used for both the 20 frequency (Veers $_{\text{red}}$) and the phase increment (Veers $_{\text{red, }\Delta\theta}$) implementations. Hence the $T = 6,000$ s signal repeats after 600 s. The spectrum is binned into discrete bins of frequencies $f_m$ equal to the

[Figure]

**Figure 9.** Wind speed cross power spectral density for three point pairs from different models, together with the analytic results (Eq. 9) and the prescribed Kaimal auto-spectrum. Left: the base point, and its closest neighbor. Middle: the base point and a point far away. And right: a point pair close together, but far away from base point.

logarithmically spaced frequencies initially used to generate the wind speed time series.

The wind speed auto-spectrum is included in Fig. 2. By definition (Eq. 8), the reduced order model produces the prescribed (auto-) spectrum exactly. Fig. 9 shows a comparison of the cross-spectra estimates for different point pairs obtained through

5  Welch's periodogram method (Welch, 1967) employed on the full 6,000 s samples with no extra windowing. We study the base point, and its closest neighbor ($P_1$-$P_6$); the base point and a point far away ($P_1$-$P_5$); and a point pair close together, but far away from base point ($P_5$-$P_{10}$). For reference, the prescribed Kaimal spectrum $S$ is included, as well as the analytic cross-spectrum obtained by:

$$S_{ij} = \underline{Coh}\underline{coh}\sqrt{S_{ii}S_{jj}} \tag{9}$$

10   with the (auto-) spectra $S_{ii} = S$ and the coherence function  $\underline{coh}$ as defined by the standard IEC 61400-1, Ed. 3 (2005).

Again, the phase increment model (Veers $_{\text{red}, \Delta\theta}$) in all cases reproduces the analytic spectrum well with only 20 random variables. This time, however, the  *TurbSim* results do not match as well. The reason is that  *TurbSim* chooses

15  the lowest frequency $f_1$ and the frequency bin width $\Delta f$ such that $\Delta f = f_1 = 1/T$, and thus uses a wider frequency band for the first bin compared to our logarithmically spaced bins. When re-binning to the logarithmic range this results in excess power (and an artificial peak) in the first bin and hence less power in higher frequency bins. Note, however, that this is an artifact of the discrete spectrum and the frequency binning, and not a discrepancy in the underlying data.

[Figure]

**Figure 10.** Blade thrust load probability distribution from  model based on wind fields generated with either  TurbSim or from the reduced order Veers model with constant phase increments.

**3.4 Outlook: wind turbine rotor blade loads**

To further assess the validity of the reduced order wind model, loads were calculated for one single blade on a three bladed $R = 35$ m diameter wind turbine rotor spinning at a tip speed ratio $\lambda = 6.1$. Loads were obtained at $\Delta t = 0.1$ s time steps through a simple blade element momentum model supplied with wind generated either from  TurbSim, or from our reduced model with fixed phase increments (Veers $_{red, \Delta\theta}$) on a $15 \times 15$ grid of data points over the rotor disc. The hub height is set to $h_{hub} = 90$ m, with the hub height mean wind speed $\bar{u} = 12$ m/s, power law wind shear with power law exponent $a = 0.2$ (according to Jonkman and Kilcher (2012)), and IEC normal turbulence model, class A (IEC 61400-1, Ed. 3, 2005). Fig. 10 shows the probability distribution $p(T)$ of thrust loads $T$ on one blade calculated from 100 realizations of a 600 s wind field.  TurbSim used the full set of roughly 3,000 frequencies at each of the $15 \times 15$ grid points. The reduced order model, on the other hand, relied on only 20 frequencies with all 100 realizations generated one set of fixed phase increments. It can be seen, that the reduced order model, although relying on significantly fewer random variables ($N_R = 6.75 \cdot 10^5$ versus $N_R = 20$ for each realization, see Tab. 1), produces almost the same load probability distribution. In a forthcoming publication (Fluck and Crawford, 2017b) we will use the reduced order wind model presented here to develop a stochastic wind turbine blade load model (BEM), This publication will also discuss these blade load PDFs in greater detail.

**3.5 Discussion**

As shown  in this section the phase increments wind model presented in section 2.2 can reproduce important statistics (both of a wind field, as well as for resulting wind loads) with the  accuracy comparable to the full model. At the same time, the phase increments model requires significantly less random variables. As indicated by Fig. 6 the phase increments model does *not* produce identical spatial structures with each realization, even thought a large

part of the spatial randomness is neglected  in Eq. 8. This further illustrates the method's ability to retain important stochastic information. Like Veer's model, our method relies on a spectral representation of a random process. The wind field is generated through an inverse Fourier transform with random phase increments. Thus the underlying process is (per definition) assumed to be stationary. As with any spectral model, singular transient (and as such deterministic) events like fronts or downbursts, can consequently not be modeled directly. It might be possible to model these events through superposition of a discrete deterministic event with our random model. This could be similar to Chay et al. (2006), who superposes a deterministic downburst profile over an ARMA model for the turbulent fluctuations. However, it is our goal to derive a formulation for wind as input to stochastic models. These models will have difficulties dealing with singular deterministic events themselves. Hence we doubt if progressing along this way would be beneficial; Nonetheless, this could obviously be the subject of a future study.

[revised manuscript text omitted]

Le Maître, O. P. and Knio, O. M.: Spectral methods for uncertainty quantification: with applications to computational fluid dynamics, Springer,, doi:10.1007/978-90-481-3520-2, 2010.

5 Majda, A. J. and Branicki, M.: Lessons in uncertainty quantification for turbulent dynamical systems, Discrete Cont. Dyn. Systems, 32, doi:10.3934/dcds.2012.32.3133, 2012.

Morales, A., Wächter, M., and Peinke, J.: Characterization of wind turbulence by higher-order statistics, Wind Energy, 15, 391–406, doi:10.1002/we.478, 2012.

Moriarty, P.: Database for validation of design load extrapolation techniques, Wind Energy, 11, 559–576, doi:10.1002/we.305, 2008.

10 Mücke, T., Kleinhans, D., and Peinke, J.: Atmospheric turbulence and its influence on the alternating loads on wind turbines, Wind Energy, 14, 301–316, doi:10.1002/we.422, 2011.

Najm, H. N.: Uncertainty quantification and polynomial chaos techniques in computational fluid dynamics, Annual Review of Fluid Mechanics, 41, 35–52, doi:10.1146/annurev.fluid.010908.165248, 2009.

Nielsen, M., Larsen, G. C., Mann, J., Ott, S., Hansen, K. S., and Pedersen, B. J.: Wind simulation for extreme and fatigue loads, Technical
15 Report Risø-R-1437 (EN), Risø, 2004.

Nielsen, M., Larsen, G. C., and Hansen, K. S.: Simulation of inhomogeneous, non-stationary and non-Gaussian turbulent winds, in: Journal of Physics: Conference Series, IOP Publishing, doi:10.1088/1742-6596/75/1/012060, 2007.

Padrón, A. S., Stanley, A. P. J., Thomas, J. J., Alonso, J. J., and Ning, A.: Polynomial chaos for the computation of annual energy production in wind farm layout optimization, Journal of Physics: Conference Series, 753, 032 021, 2016.

20 Park, J., Manuel, L., and Basu, S.: Toward isolation of salient features in stable boundary layer wind fields that influence loads on wind turbines, Energies, 8, 2977–3012, doi:10.3390/en8042977, 2015.

Sudret, B.: Uncertainty propagation and sensitivity analysis in mechanical models–Contributions to structural reliability and stochastic spectral methods, Habilitation a diriger des recherches, Université Blaise Pascal, Clermont-Ferrand, France, 2007.

Sullivan, T. J.: Introduction to uncertainty quantification, Springer, Switzerland, 2015.

25 Tibaldi, C., Henriksen, L. C., and Bak, C.: Investigation of the dependency of wind turbine loads on the simulation time, in: Proceedings of EWEA 2014, Barcelona, Spain, European Wind Energy Association (EWEA), 2014.

Veers, P. S.: Three-dimensional wind simulation, Technical Report SAND88–0152, UC–261, Sandia National Labs, 1988.

Welch, P.: The use of fast Fourier transform for the estimation of power spectra: A method based on time averaging over short, modified periodograms, IEEE Transactions on Audio and Electroacoustics, 15, 70–73, doi:10.1109/TAU.1967.1161901, 1967.

30 Zwick, D. and Muskulus, M.: The simulation error caused by input loading variability in offshore wind turbine structural analysis, Wind Energy, 18, 1421–1432, doi:10.1002/we.1767, 2015.